



# Wintertime Saharan dust transport towards the Caribbean: airborne lidar observations during EUREC[4]A

Manuel Gutleben[1,2], Silke Groß[1], Christian Heske[1], and Martin Wirth[1]

[1]Deutsches Zentrum für Luft- und Raumfahrt, Institut für Physik der Atmosphäre, Oberpfaffenhofen, Germany
[2]Ludwig-Maximilians-Universität, Meteorologisches Institut, Munich, Germany

**Correspondence:** MANUEL GUTLEBEN (manuel.gutleben@dlr.de)

**Abstract.** Wintertime Saharan dust plumes in the vicinity of Barbados are investigated by means of airborne lidar measurements. The measurements were conducted in the framework of the EUREC[4]A field experiment (Elucidating the Role of Cloud-Circulation Coupling in Climate) upstream the Caribbean island in January/February 2020. The combination of the water vapor differential absorption and high spectral resolution lidar techniques together with dropsonde measurements aboard the

German HALO (High Altitude and Long-Range) research aircraft enable a detailed vertical and horizontal characterization of the measured dust plumes. In contrast to summertime dust transport, mineral dust aerosols were transported at lower altitudes and were always located below 3.5 km. Calculated backward trajectories affirm that the dust-laden layers have been transported in nearly constant low-level altitude over the North Atlantic Ocean. Only mixtures of dust-particles with other aerosol species, i.e. biomass burning aerosol from fires in West Africa and marine aerosol, were detected by the lidar. No pure mineral dust

regimes were observed. Additionally, all the dust-laden airmasses that were observed during EUREC[4]A came along with enhanced water vapor concentrations compared to the free atmosphere above. Such enhancements have already been observed during summertime and were found to have a great impact on radiative transfer and atmospheric stability.

## 1   Introduction

Mineral dust aerosol is known to be a major contributor to the Earth's aerosol mass burden (Cakmur et al., 2006) and is

estimated to contribute between 25 % and 30 % to the total aerosol optical depth (Tegen et al., 1997; Kinne et al., 2006). The greatest source region of mineral dust is the Saharan desert and its arid surrounding landscapes. From a comparison of 15 global aerosol models, Huneeus et al. (2011) derived annual dust emissions from North Africa that range from 400 to 2200 $\mathrm{Tg\,a^{-1}}$. This makes up roughly 50 % of the total global annual dust emission (1000 to 4000 $\mathrm{Tg\,a^{-1}}$).

Once injected into the atmosphere, Saharan dust particles can be transported far away from their origin. While only 15 % of

the total emitted dust load is transported north- and eastwards towards the Mediterranean and the Middle East (e.g. Shao et al., 2011), almost 85 % of the dust burden gets carried south- and westwards over the Atlantic Ocean. Transatlantic transportation routes vary throughout the year and depend on the large-scale synoptic situation (e.g. Ben-Ami et al., 2009; Tsamalis et al., 2013). During the winter months the Harmattan carries Saharan dust particles at low levels (<3.5 km) southwestwards towards the Gulf of Guinea (Schepanski et al., 2009). Due to the southernmost position of Intertropical Convergence Zone (ITC) during





this season, dust particles are then frequently transported towards the South American continent and the Amazon region (e.g.
Yu et al., 2015). During boreal summer, however, strong solar insolation over the Sahara causes a deep boundary layer (e.g.
Messager et al., 2009) which can reach up to 6 km altitude (≈500 hPa). This allows dust particles to be mixed upward to higher
altitudes where they get caught by the trade winds. Due to the northward shift of the ITC during boreal summer, they then get
carried westwards as far as the Caribbean as well as Central and North America along a more northerly transportation route
(e.g. Prospero and Carlson, 1972; Prospero et al., 2010).

Summertime dust advection can frequently be observed in the Caribbean. During that time of the year the dust particles are
usually advected in elevated and decoupled layers - so called Saharan air layers (SALs; Prospero and Carlson, 1972; Carlson
and Prospero, 1972; Prospero et al., 2021). Wintertime dust plumes, however, reach the Caribbean less frequently and only in
connection with favorable synoptic situations. They are mostly transported at lower levels and are sometimes mixed with other
aerosol types like biomass burning aerosols or marine aerosols in the marine boundary layer (MBL; Chiapello et al., 1995;
Ben-Ami et al., 2009; Groß et al., 2011).

Recent studies have shown, that summertime SALs may come along with enhanced concentrations of water vapor compared
to the surrounding dry free troposphere (Gutleben et al., 2019b, 2020; Ryder, 2021). It was shown that not the aerosol but the
water vapor inside the SALs is the dominating driver for net radiative heating during transatlantic dust transport. In this way,
water vapor associated with dust layers has the potential to modify the atmospheric stability and consequently to impact the
evolution as well as macrophysical properties of shallow marine trade wind clouds.

In addition to that, the transportation of dust particles at low atmospheric levels within/atop of the MBL during wintertime
implies aerosol radiative effects. The particles may not only have a direct effect on the evolution of shallow marine clouds via
absorption and scattering, but also have an indirect effect as dust particles can act as cloud condensation and ice nucleating
particles (Karydis et al., 2011; Bègue et al., 2015; DeMott et al., 2015; Boose et al., 2016). However, before an investigation of
these effects can be performed, wintertime transatlantic dust transport towards the Caribbean has to be investigated in detail,
which is the focus of this study.

Up to now, the concurrence of transatlantic Saharan dust and water vapor transport during boreal winter has never been
studied. In early 2020, extensive wintertime mineral dust plumes could be observed upstream the Caribbean island of Barbados
during the EUREC⁴A field experiment (Elucidating the Role of Cloud-Circulation Coupling in Climate; Stevens et al., 2021).
Several research flights with the High Altitude and Long Range Research Aircraft (HALO; Krautstrunk and Giez, 2012) were
carried out over these regions and an extensive lidar and dropsonde data set was collected. This unique data set now enables a
detailed investigation of the plumes.

This paper is structured as follows. In section 2 an overview of the EUREC⁴A project as well as of the performed research
flights is given. Moreover, the capabilities and features of the airborne lidar system WALES (Water Vapor Lidar Experiment
in Space; Wirth et al., 2009) are introduced. A detailed analysis of the observed Saharan dust events by means of airborne
lidar and dropsonde measurements is presented in section 3. Finally, in section 4, the results are discussed and the study is
concluded.





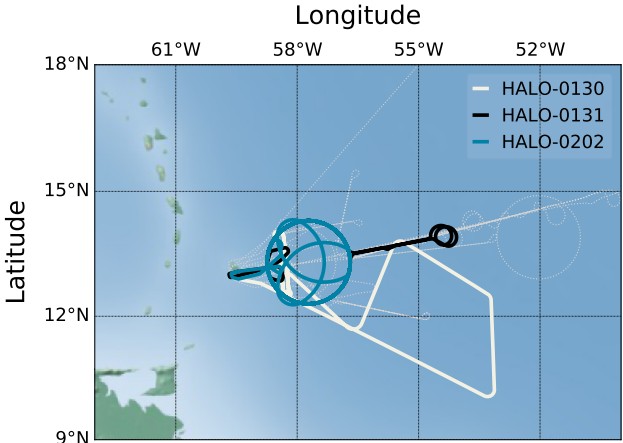

**Figure 1.** The HALO flight tracks during EUREC[4]A. Flights over long-range-transported Saharan mineral dust are colored. The thin dashed white lines indicate the remaining dust-free research flight-tracks.

## 2 Methods

### 2.1 HALO research flights during EUREC[4]A

The EUREC[4]A field campaign aimed at investigating the driving factors for the evolution of trade-wind cumulus clouds in the winter trades (Stevens et al., 2021). As one of several employed research platforms during EUREC[4]A, the German high-flying research aircraft HALO conducted remote sensing measurements east of the Caribbean island of Barbados (Konow et al., 2021). In the period from 18 January to 18 February 2020, the modified Gulfstream G550 research aircraft performed a total of 15 scientific flights (13 local flights from and to Barbados and two transfer flights from and to Germany). An overview of the HALO flight tracks is shown in Figure 1. The circular flight patterns of the research flights were flown for dropsonde-based analyses of the respective prevailing large-scale divergences (Bony and Stevens, 2019).

On three measurement days, Saharan mineral dust plumes were advected to the research area, i.e. on 30 Jan 2020 during HALO-0130, on 31 Jan 2020 during HALO-0131 and on 2 Feb 2020 during HALO-0202. Collected airborne lidar data sets during measurement flights on these days enable a characterization of winter-time dust transport, although the flights tracks have not been specifically designed for dust observations. The enhanced total column aerosol optical depth (AOD) over the dust-affected research areas was also captured by MODIS (Moderate-resolution Imaging Spectroradiometer) and is shown in Figure 2. While on 2 February the AOD in the research area ranged from 0.2 to 0.4, it was higher during the two other days and took maximum values greater than 0.6. The African origin of the aerosol plumes is also verified by calculated backward trajectories (Figure 4) using the Hybrid Single Particle Lagrangian Integrated Trajectory model (HYSPLIT; Stein et al., 2015).



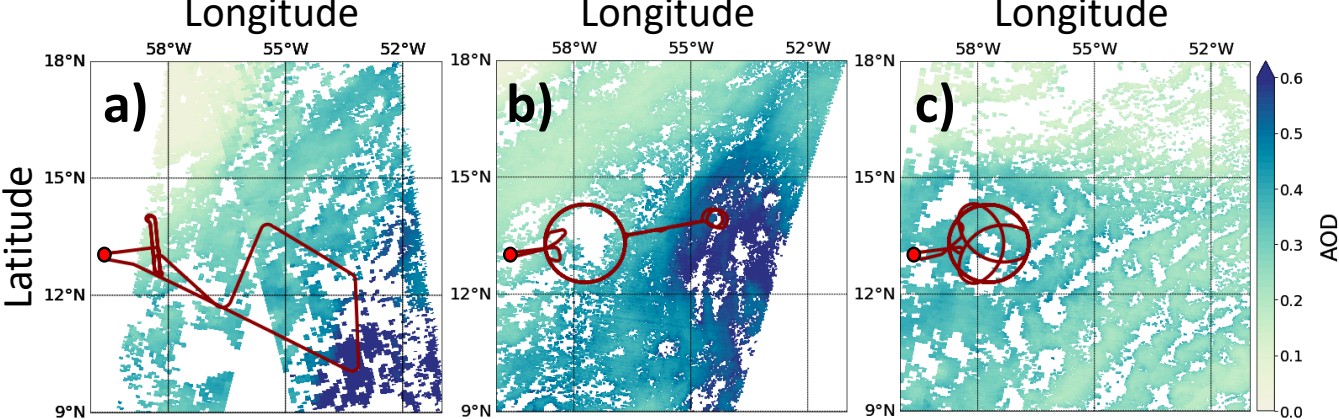

**Figure 2.** The HALO flight-tracks of the three research flights over long-range-transported Saharan dust aerosol on 30 Jan 2020 **(a)**, 31 Jan 2020 **(b)** and 2 Feb 2020 **(c)** atop of MODIS (Moderate-resolution Imaging Spectroradiometer) satellite imagery of aerosol optical depth (AOD). All MODIS images were taken between 14 and 18 UTC. The Grantley Adams International Airport on Barbados is marked with red dots.

In total, approximately 18.5 hours of lidar data could be collected during the three flights. Additionally, a total of 164 dropsondes were launched (Vaisala RD-41; George et al., 2021; Vaisala, 2020). 157 of them worked properly and collected thermodynamic data from aircraft to ground level.

## 2.2 The WALES lidar instrument aboard HALO

During EUREC$^4$A the WALES lidar instrument (Wirth et al., 2009) was operated aboard HALO. It is an airborne water vapor differential absorption lidar (DIAL) system with depolarization and high spectral resolution lidar (HSRL; Esselborn et al., 2008) capabilities.

The DIAL module operates at four wavelengths (three online and one offline wavelength) around the $H_2O$- absorption band at 935 nm. This setup allows for measurements of water vapor mass mixing ratios ($r_m$) that cover the whole extent from aircraft to ground level. Due to its high pulse-repetition rate of 0.01 s between online and offline pulses, it enables horizontally and vertically highly resolved measurements of $r_m$ with relative uncertainties of less than 5 % (Kiemle et al., 2008). To obtain the high pulse repetition rate, two Q-switched Nd:YAG (Neodymium-doped Yttrium Aluminum Garnet; $Nd : Y_3Al_5O_{12}$) ring lasers, that generate pulses at 1064 nm wavelength, are operated temporally interleaved. The required frequencies for DIAL measurements are generated via frequency-doubling of parts of the emitted pulses and subsequent pumping of injection-seeded optical parametric oscillators (Mahnke et al., 2007).





**Table 1.** Overview of the conducted research flights over Saharan dust aerosol during EUREC[4]A in 2020 including dates, times of take-off and landing, total duration as well as number of launched dropsondes (times given in UTC - note: Atlantic Standard Time = UTC-4).

| ID | Date | Take-off | Landing | Duration | Sondes |
|----|------|----------|---------|----------|--------|
| 0130 | 30 Jan | 11:19 | 15:09 | 04:50 h | 4 |
| 0131 | 31 Jan | 14:07 | 00:10 (+1 d) | 10:03 h | 74 |
| 0202 | 02 Jan | 11:28 | 20:15 | 08:47 h | 89 |

In addition to the DIAL capability, the integrated HSRL-module and depolarization sensitive channels allow for highly resolved measurements of particle linear depolarization ratios ($\delta_{p(532)}$), backscatter ratios ($R_{532} = 1 + \beta_{p(532)}/\beta_{m(532)}$; with $\beta_{p(532)}$ and $\beta_{m(532)}$ being the particle backscatter and molecular backscatter coefficients), particle extinction coefficients ($\alpha_{p(532)}$) as well as the lidar ratios ($S$) at 532 nm wavelength. Relative uncertainties in measurements of $\delta_{p(532)}$, $R_{532}$ and $\alpha_{p(532)}$ are 10 % to 16 %, 5 % and 10 % to 20 %, respectively (Esselborn et al., 2008).

WALES data is spatially and temporally averaged to improve the signal to noise ratio. At typical aircraft speed of 200 m s$^{-1}$, the horizontal resolutions amount to 3000 m (DIAL) and 200 m (HSRL). The vertical resolution is 15 m.

## 3 Results

### 3.1 Saharan dust research flights during EUREC[4]A

Transported Saharan mineral dust was present in the research area on three measurement days. Times of take-off and landing of the flights conducted on these days, as well as the number of launched dropsondes are summarized in Table 1. In the following, a detailed overview of the measurements during these three flights is given.

a. *HALO-0130 on 30 January 2020:*

An objective of this relatively short research flight was an under-flight of the GPM (Global Precipitation Measurement Mission; Hou et al., 2014) satellite for concurrent airborne and spaceborne radar and radiometer measurements of subtropical clouds. For this purpose the flight track was designed to lead to a region in the Southeast of Barbados, where long-range-transported Saharan dust aerosol prevailed. The time-height cross section of $\delta_{p(532)}$ highlights a highly depolarizing aerosol regime which ranged from approximately 0.7 km to 3.5 km altitude during the whole flight (Figure 3 (a)). Pure and non-mixed long-range-transported Saharan mineral dust aerosol usually takes typical values around 30 % (Freudenthaler et al., 2009; Groß et al., 2013). This value does not change with transatlantic transportation (Groß et al., 2015). However, $\delta_{p(532)}$ between 0.7 km to 3.5 km altitude took slightly smaller values that ranged from 15 % to 30 %. Depolarization ratios in this range are typical for aerosol regimes where mineral dust particles are mixed with less depolarizing particles, i.e. marine aerosol or biomass burning aerosol. Mixing of different aerosol types is also evident

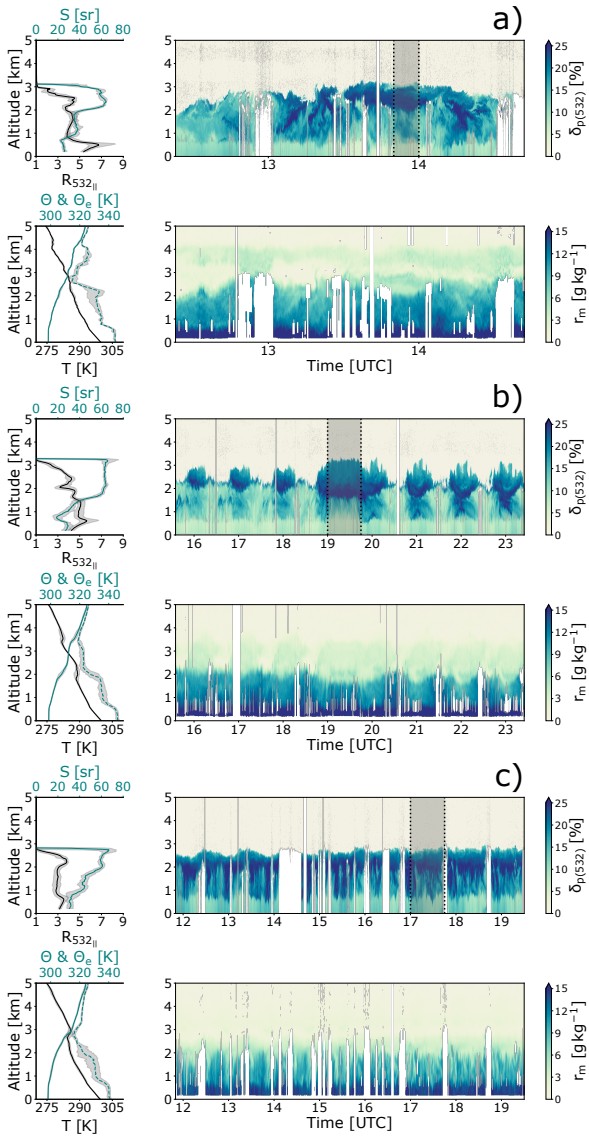

**Figure 3.** Overview of measured profiles during the HALO-0130 **(a)**, HALO-0131 **(b)** and HALO-0202 **(c)** research flights as measured by lidar and dropsondes. The left panels show median profiles of 532 nm backscatter ratio ($R_{532_{||}}$, black), lidar ratio (S, teal), temperature (T, black), equivalent potential temperature ($\Theta_e$, teal, dashed) and potential temperature ($\Theta$, teal, solid) during the respective flights. Shaded regions indicate the inter-quartile region. The right panels show the measured profiles of 532 nm particle linear depolarization ratio ($\delta_{p(532)}$) and water vapor mass mixing ratio ($r_m$). Since measurements of the WALES DIAL are masked at altitudes <200 m (contamination from surface echoes), no water vapor measurements near ground level are available.





in derived lidar ratios at 532 nm. From ground level to approximately 0.7 km altitude they took values around 30 sr. In intermediate altitudes (0.7 km to 2.0 km) where $\delta_{p(532)}$ took values around 15 % lidar ratios were around 40 sr. At the top levels of the depolarizing layer (2.0 km to 3.5 km) where $\delta_{p(532)}$ was highest, the derived lidar ratios were also highest and took values around 60 sr. This is a clear indication that at the top levels of the areosol layer highly absorbing biomass burning aerosols ($S$ of 63±7) are mixed with mineral dust particles (typical $S$ of 50±4). On the other hand, at the lower levels of the aerosol layer, values of $\delta_{p(532)}$ and $S$ point towards a mixture of marine aerosols ($S$ of 18±5) with dust particles (Ben-Ami et al., 2009; Groß et al., 2011).

Ensembles of 7-day backward trajectories that are calculated utilizing HYSPLIT show that the sampled dust-containing aerosol layer traveled for one week at a nearly constant altitude from Africa towards the Caribbean (Figure 4 (a)). Thermal anomalies derived from VIIRS (Visible Infrared Imaging Radiometer Suite) indicate strong activity of fires in West Africa. The low-latitude transportation route of the aerosol layer (5°N to 15°N) together with these fires favored the entrainment of biomass-burning aerosols into the boundary layer. A detailed discussion and analysis of the observed aerosol mixtures is given in subsection 3.2.

During HALO-0130 air masses at low atmospheric levels (trajectories starting at 0.4 km altitude) took a more northerly route over the North Atlantic Ocean (10°N to 15°N) than the dust-laden air masses at 1.2 km and 2.0 km. This could explain why the MBL at lowermost altitudes (<0.7 km) is characterized by relatively small $\delta_{p(532)}$ and $S$ compared to the overlying atmosphere. The altitude of ~0.7 km altitude marked a transition inside the MBL. The MBL changed from a moist and neutrally stratified convective boundary layer near the surface ($r_m \sim 18\,\mathrm{g\,kg^{-1}}$; $\Theta$ = const.) to a dust-laden, drier and more stable layer with lower wind speeds ($r_m$ = 4-10 $\mathrm{g\,kg^{-1}}$; $d\Theta/dz \geq 0$). This layer is then capped at approximately 2.5 km altitude by the trade wind inversion (TWI). The TWI was characterized by a hydrolapse, gradients in wind speed as well as temperature and the transition from an aerosol-laden to an aerosol-free atmosphere.

b. *HALO-0131 on 31 January 2020:*
HALO's second dust-flight aimed at measurements of large-scale divergence from dropsondes. This is why a total of seven circles were flown upstream of Barbados and dropsondes were launched at a very high rate. After 3.5 circles a short excursion towards the Northwest Tropical Atlantic Station - a meteorological buoy - was performed. The sampled atmosphere below the circles was characterized by an almost dust-free region in the Northwest and a dusty regime in the Southeast (see Figure 2 (b)). As a result the lidar depolarization data shows recurring features (Figure 3 (b)). Highly depolarizing and dust-laden regions were observed between 0.7 km and 3 km altitude (15 %<$\delta_{p(532)}$<30 %), with greatest depolarization ratios in 2 km altitude. In these altitudes the lidar ratio $S$ took values around 60 sr, indicating the presence of both mineral dust and biomass burning aerosol. At lower levels (0.0 km to 0.7 km) $S$ was smaller and around 20 sr. This points towards an additional contribution of marine aerosol in these altitudes.

Similar to HALO-0130, backward trajectories with starting points at low altitudes originate from more northern latitudes than the trajectories starting from altitudes affected by long-range-transported Saharan mineral dust, i.e. at 1.2 km and 2.0 km altitude. The low latitudes together with low transportation altitudes can again explain that not only mineral dust,





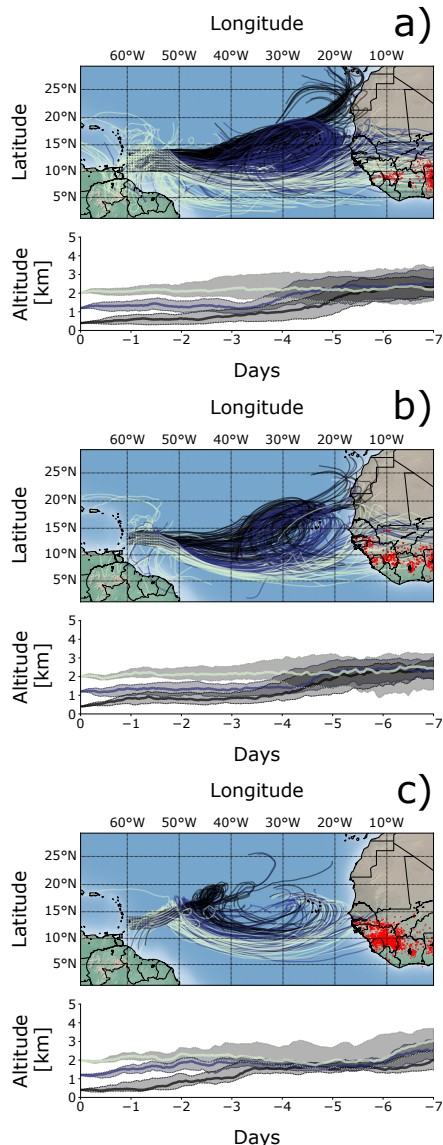

**Figure 4.** The 7- d backward trajectories of the air masses that were sampled during the HALO-0130 **(a)**, HALO-0131 **(b)** and HALO-0202 **(c)** research flights. Trajectories are indicated for starting points in 0.4 km (black), 1.2 km (blue) and 2.0 km (light-green) altitude that are regularly distributed over the research areas (0.5 ° spacing) and are calculated using the Hybrid Single Particle Lagrangian Integrated Trajectory model HYSPLIT. Red markers indicate detected thermal anomalies related to active fires detected by VIIRS (Visible Infrared Imaging Radiometer Suite) 7 days before the respective research flights. The lower panels show the median altitude of the backward trajectories. Shaded regions indicate the inter-quantile region.





but also biomass burning aerosol from West African fires, marine aerosol and Saharan mineral dust, were observed. This
is substantiated by the evident forest fires detected from VIIRS thermal anomalies. The MBL was characterized by a
moist and well-mixed convective layer from surface to approximately 0.7 km altitude ($r_m \approx 18\,\mathrm{g\,kg^{-1}}$; $\Theta$ = const.) and
a dryer, calmer and statically more stable layer from 0.7 km to 2.2 km which was capped by the TWI. A relatively moist
and dust-containing aerosol layer was observed atop of the MBL. It was vertically extending for roughly one kilometer
and showed water vapor mass mixing ratios of approximately $4\,\mathrm{g\,kg^{-1}}$.

c. *HALO-0202 on 2 February 2020:*

Circular flight patterns for large-scale divergence measurements have also been the focus of this research flight. For an
additional characterization of the cloud and aerosol scene in the interior of the circles a clover-pattern was flown. As a
result WALES lidar measurements during this research flight show recurring features (Figure 3 (c)).

The MBL was again characterized by a well-mixed convective layer from surface to approximately 0.7 km altitude ($\Theta$
= const.). However, water vapor mixing ratios were slightly smaller compared to the other two flights ($\sim 16\,\mathrm{g\,kg^{-1}}$) .
The MBL was capped at approximately 2.5 km altitude by a pronounced TWI. Depolarizing aerosol was found along the
whole vertical extent of the MBL. However, $\delta_{p(532)}$-values were highest at the very top of the MBL ($\sim$30 % at 2.3 km
altitude). Lidar ratios again indicate that the aerosol layer was characterized by a particle mixture of mineral dust aerosol,
biomass-burning aerosol and marine aerosol. $S$ took highest values inside the aerosol layer atop of the MBL ($\sim$60 sr)
and was lowest near ground level ($\sim$30 sr). This indicates that the contribution of biomass burning aerosol to the aerosol
mixture was greatest at the top levels and marine aerosol mostly contributed to the aerosol mixture at lower levels.
Regions inside the MBL that were impacted by this aerosol mixture came along with reduced atmospheric moisture
content and were characterized by small water vapor mixing ratios that ranged from $5\,\mathrm{g\,kg^{-1}}$ to $10\,\mathrm{g\,kg^{-1}}$.

7-d backward trajectories for this research flight again indicate that the dust-laden air masses in 1.2 km and 2.0 km
altitude traveled at nearly constant altitude and that they took a southerly route that favored mixing processes with
biomass burning aerosols and marine aerosols.

## 3.2 Aerosol and water vapor composition of the observed dust layers

Measurements of lidar ratios and particle linear depolarization ratios during the three research flights over transported Saharan
mineral dust indicated that dust particles have been mixed together with biomass burning and marine aerosols. An aerosol
regime containing only mineral dust particles was never observed during EUREC⁴A. To investigate these mixed regimes in
more detail, the contribution of Saharan mineral dust aerosol to the aerosol mixture is calculated. The well-established aerosol-
separation technique for two-component aerosol mixtures shown by Tesche et al. (2009) and Groß et al. (2011) allows for the
calculation of the contribution (in percent) of mineral dust to both the measured particle linear depolarization ratio $\delta_{p(532)}$ and
the measured lidar ratio $S$.





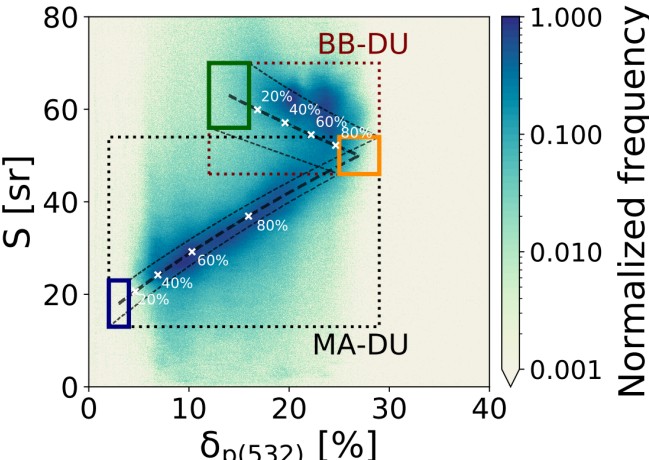

**Figure 5.** Two-dimensional histogram showing the joint distribution of measured particle linear depolarization ($\delta_{p(532)}$) and lidar ratios ($S$) during the three research flights over long-range-transported Saharan mineral dust. Solid rectangles mark the typical positions of the pure aerosol regimes: marine aerosol (blue), transported Saharan mineral dust (orange) and biomass burning aerosol (green). The dashed rectangles indicate the mixing regimes of biomass burning aerosols with transported Saharan dust aerosols (BB-DU, maroon) as well as of marine aerosols with transported Saharan dust aerosols (MA-DU, blue). Grey lines indicate the calculated mixing lines (dashed: mean, solid: ± standard deviation) of the two aerosol mixtures MA-DU and BB-DU. White crosses and percentages indicate the contribution of dust to $\delta_{p(532)}$ of the respective two-component mixtures. The histogram is normalized to the maximum bin count.

The percentage that Saharan mineral dust is contributing to a measured particle linear depolarization ratio of a two-component aerosol mixture $\delta_{p(532)}$ with biomass burning aerosol or marine aerosol is calculated as follows,

$$[\%_\delta] = \frac{D_B}{D_B + D_A},$$ (1)

with the coefficients $D_A$ and $D_B$,

$$D_A := \frac{\delta_{p(532),DU} - \delta_{p(532)}}{S_{DU}(1 + \delta_{p(532),DU})};$$ (2)

$$D_B := \frac{\delta_{p(532)} - \delta_{p(532),BB|MA}}{S_{BB|MA}(1 + \delta_{p(532),BB|MA})}.$$ (3)

Here, $S_{DU}$ and $S_{BB|MA}$ are the known lidar ratios of Saharan mineral dust (50±4) as well as of biomass burning aerosols (63±7) and marine aerosol (18±5) (mean values of observations by Groß et al., 2013). $\delta_{p(532),DU}$ and $\delta_{p(532),BB|MA}$ are the corresponding known particle linear depolarization ratios (Saharan mineral dust: 27±2 %; biomass burning aerosol: 14±2 %; marine aerosol: 3±1 %). For $S$ the percentage is calculated using,

$$[\%_S] = \frac{S_{BB|MA}(S_{DU} - S)}{S(S_{BB|MA} - S_{DU})}.$$ (4)





Using these equations, one can derive mixing lines between the pure Saharan mineral dust aerosol regime and the marine and biomass burning aerosol regime in a two-dimensional space of $S$ and $\delta_{p(532)}$ (Figure 5). The joint histogram of the lidar measurements during the three flights over long-range-transported Saharan mineral dust aerosol fits well to the calculated mixing
lines. Maximum frequencies of measurements correlate very well with the mixing lines between mineral dust and biomass burning aerosol as well as mineral dust and marine aerosol derived from the calculations. The histogram also shows that biomass burning aerosols are not mixed to low atmospheric levels as this would cause a shift of the mixing line between marine aerosol and dust aerosol towards higher lidar ratios. Mineral dust contributions between 40 % and 80 % were observed most frequently in mixtures of marine aerosol with dust aerosol (MA-DU). At higher atmospheric levels mineral dust contributed between 20 %
and 80 % to the mixture of biomass burning aerosol and dust aerosol (BB-DU).

Gutleben et al. (2019a, 2020) observed enhanced water vapor concentrations of approximately $4\,\mathrm{g\,kg^{-1}}$ in elevated and long-range-transported SALs compared to the surrounding atmosphere during the boreal summer months. One objective of the lidar measurements during EUREC[4]A is to answer the question whether this is also the case for transatlantic dust transport during the boreal winter season.

When looking at the individual lidar curtains of the three research flights in Figure 3 one can already identify the slightly enhanced water vapor mass concentrations compared to the dry free troposphere in regions of highest $\delta_{p(532)}$. Measurements during HALO-0131 and HALO-0202 highlight that the regions of maximum $\delta_{p(532)}$ - which are associated with a great mineral dust loading - on top of the MBLs are not dry, but carry water vapor around $4\,\mathrm{g\,kg^{-1}}$.

This can also be seen in the joint distributions of $r_m$ with $\delta_{p(532)}$ and $S$ (see Figure 6). It indicates that regions affected by
mineral dust are not completely dry, but always come along with enhanced water vapor concentrations. Only in mixed regions, where biomass-burning aerosol dominates the dust aerosol ($\delta_{p(532)} \approx 20\,\%$; $S \approx 60\,\mathrm{sr}$), water vapor mixing ratios drop to values smaller $4\,\mathrm{g\,kg^{-1}}$. In the lower MBL, where a mixture of mineral dust with marine aerosol predominates, water vapor concentrations are high in general and take values between $8\,\mathrm{g\,kg^{-1}}$ and $18\,\mathrm{g\,kg^{-1}}$.

## 4 Discussion and Conclusions

While during the summer months Saharan dust particles are predominantly transported westwards in SALs at altitudes as great as 6 km (e.g. Prospero and Carlson, 1972; Prospero et al., 2010), Saharan dust transport in the winter months happens at lower atmospheric levels (Chiapello et al., 1995). This agrees well with the findings in this study, as mineral dust particles were never observed in altitudes higher than $\sim 3.5$ km. Low level transport also favors mixing processes of mineral dust particles with other aerosol species like biomass burning aerosol or marine aerosol. As a consequence, pure dust aerosol regimes were
never observed during EUREC[4]A and mineral dust particles could only be observed in mixed aerosol regimes. At lowermost altitudes inside the MBL the dust particles were predominantly mixed with marine aerosol. Above the MBL in altitudes from 2 km to 3.5 km the dust particles were mixed with biomass burning aerosol from fires in West Africa.

During their travel over the Atlantic Ocean these dusty aerosol regimes can impact the Earth's radiation budget by directly interacting with radiation via absorption and scattering, by changing cloud microphysical properties or by modifying the





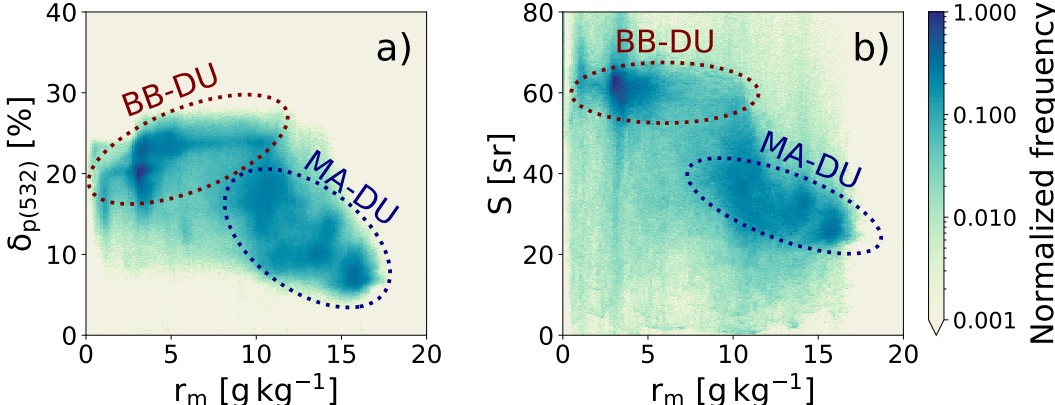

**Figure 6.** Two-dimensional histograms showing the joint frequency distribution of measured particle linear depolarization $\delta_{p(532)}$ (**a**) and lidar ratios $S$ (**b**) with simultaneously measured water vapor mass mixing ratios $r_m$ during the three research flights over long-range-transported Saharan mineral dust. The dashed ellipses indicate an outline of the mixing regimes of biomass burning aerosols with transported Saharan dust aerosols (BB-DU, maroon) as well as of marine aerosols with transported Saharan dust aerosols (MA-DU, blue). The histograms are normalized to their respective maximum bin count.

atmospheric stratification. However, up to now the radiative impact of long-range Saharan mineral dust transport over the Atlantic Ocean during boreal winter has never been investigated in detail. Needed highly resolved observations of the vertical and horizontal aerosol distribution were missing. For summertime transport Gutleben et al. (2019b, 2020) highlighted that not the aerosol, but the water vapor embedded in long-range-transported SALs is the dominant driver for radiative heating and the subsequent modification of the atmospheric stability. They used a radiative transfer model together with airborne lidar data

collected during NARVAL-II (Next-generation aircraft remote sensing for validation studies II) for their calculation. During summertime the dust transport usually occurs in elevated layers. Due to this spatial separation of the different aerosol layers, the impact of enhanced concentrations of water vapor in SALs on atmospheric heating could easily be quantified. However, wintertime low-level transport - as observed during EUREC$^4$A - hampers such a separation as the water vapor concentration inside the MBL is high on principle. In addition, the indirect radiative effect of Saharan mineral dust and biomass burning

aerosols (i.e. the modification of marine cloud microphysics by aerosol particles) has to be considered in future simulations of radiative transfer in the winter season. Haarig et al. (2019) demonstrated that smoke particles play a crucial role for cloud condensation especially during boreal winter as they dominate the concentration of available cloud condensation nuclei in the mixture. Moreover, the potential of mixed dust aerosol regimes during winter-time transport for ice nucleation should be investigated in furture studies.

This study highlighted the characteristics of the observed dusty aerosol layers during EUREC$^4$A. In near future the composition of the dust layers and their radiative impact on the subtropical environment should be investigated in more detail. Finally,





it is highly recommended that future and ongoing analyses with focus on radiative transfer, cloud physics or cloud occurrence based on observations during EUREC[4]A consider the impacts of the pronounced aerosol layers described in this paper.

*Data availability.* The data used in this publication were collected in the framework of the field study EUREC[4]A and are publicly available
online in the AERIS database (https://observations.ipsl.fr/aeris/eurec4a/)

*Author contributions.* MG, SG and MW contributed in carrying out the airborne measurements during the EUREC[4]A field campaign. Initial lidar data processing was performed by MW. MG and CH performed all analytic computations and analyzed the measured data set. MG took the lead in writing the manuscript under permanent consultation with SG. All authors discussed and reviewed the findings and contributed to the final manuscript.

*Competing interests.* The authors declare that they have no conflict of interest.

*Acknowledgements.* EUREC4A is funded with support of the European Research Council (ERC), the Max Planck Society (MPG), the German Research Foundation (DFG), the German Meteorological Weather Service (DWD) and the German Aerospace Center (DLR). The authors would like to thank the staff members of the German Aerospace Center (DLR) HALO aircraft from DLR Flight Experiments for preparing and performing the successful measurement flights.



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
