# Peer review of "Wintertime Saharan dust transport towards the Caribbean - an airborne lidar case study during EUREC4A"

_Atmospheric Chemistry and Physics, 2022_

## Author Comment (AC1)

**Author's Response to the Referee Comments**

Manuel Gutleben, Silke Groß, Christian Heske and Martin Wirth

April 25, 2022

The authors would like to thank the referees very much for carefully reading the submitted manuscript and for their helpful and very valuable suggestions and feedbacks. In the following, all comments and questions will be addressed and answered. The comments are repeated and a direct response is given below. In addition, changes in the manuscript are highlighted in the appended marked-up manuscript version using blue (additions) and red (removals) colors.

**Reply to Comments of Referee #1 on 9 March 2022**

**Major Issues**

**(I) The biggest issue to me is that the authors seem to draw quite general conclusions from what really is a case study of an individual event. I'd therefore urge them to not overinterpret the findings and clearly state that these measurements - unique as they are - mark a limited sample that does not allow for drawing more general conclusions. This should also be expressed by re-categorization of the manuscript type as Measurement Report and a change of title to, e.g. Airborne lidar observations of a case of wintertime Saharan dust transport towards the Caribbean during EUREC⁴A.**

*Thank you for this very valuable comment. We agree, that this study draws conclusions from measurements during only one research campaign. It was already explicitly stated in the title of the submitted manuscript, that the paper is based on measurements during one campaign, namely EUREC⁴A. Nevertheless, we agree that it is appropriate to change the title of the manuscript to: 'Wintertime Saharan dust transport towards the Caribbean - an airborne lidar case study during EUREC⁴A', to point out that the paper is indeed a case study. Moreover, we point it out once more in the conclusion of the study. However, since we observed for the first time that enhanced concentrations of water vapor are also advected during winter-time dust transport, we follow the suggestion of Referee #2 not to change the manuscript type to a measurement report.*

**(II) Figure 1 should be omitted. The same is shown in a much better way in Figure 2.**

*Thank you for this suggestion. Please refer to the reply to comment (1) by Referee #2.*

**(III) The methods section (maybe better data and methods?) should also include the auxiliary data use in your work, i.e. MODIS, HYSPLIT, etc. Lines 173 to 193 should be moved to that section and expanded towards a discussion of typical values.**

*We followed your suggestion and moved the introduction to the aerosol separation technique to Chapter 2. We also changed the title of the chapter to 'Data and Methods'. HYSPLIT and MODIS are introduced in Chapter 'Data and Methods' in the revised manuscript. You can find all changes in the attached marked-up manuscript version.*

**(IV) Figures 3 and 4 should be split into three figures each and places at positions in the text so that the reader doesn't have to go back and forth to follow their discussion. As is, the panels in Figure 3 are too small. The last sentence in the caption of Figure 3 should be moved to the methodology section. A statement regarding the grey shaded areas should be mentioned.**

*Thank you for this valuable comment. We split up Figure 3 and Figure 4 in the revised manuscript such that the reader does not have to jump back and forth. This also increases the size of the panels in Figure 3. Furthermore, we followed your suggestion and specified that the median profiles of $R_{||}$ and $S$ refer to the grey shaded regions in the respective right panels.*

**Minor Issues**

**(1) line 15: is there an estimate of the dust contribution based on measurements?**
*We are not aware of any based estimate of annual North African dust aerosol emission, which is solely based on measurements. Hence, we prefer to stick to the estimate by Huneeus et al. (2011) which is based in a comparison of 15 global aerosol models.*

**(2) line 21: transport instead of transportation**
*We corrected that.*

**(3) line 24: the Intertropical Convergence Zone is generally referred to as ITCZ**
*We changed the abbreviation in the revised manuscript.*

**(4) line 44: cloud condensation nuclei**
*The mistake is corrected in the revised manuscript.*

**(5) line 50: this region instead of these regions**
*We corrected that.*

**(6) line 53: What plumes?**
*The mineral dust plumes mentioned in the sentences before. For a better understanding we changed the sentence to: "This unique data set now enables a detailed investigation of macrophysical properties of the observed long-range-transported dust plumes and the state of the atmosphere."*

**(7) line 70: enable a characterization of winter-time dust transport: please clearly state that you are discussing just three research flights within four days and that those flights are likely to cover the same dust event. In that context, your aim of characterizing wintertime dust transport is quite overstates what is possible with your data set.**
*We pointed that out by revising the paragraph to: "Collected airborne lidar data sets during measurement flights on these days enable a characterization of long-range-transported African dust plumes during EUREC$^4$A, although the flights tracks have not been specifically designed for dust observations. As dust aerosols could only be measured in this 4-day period, it is likely that the observed aerosol originated from the very same African dust outbreak."*

**(8) Figure 2 and related text: Please: at what wavelength of AOD measurement. Caption: one station is marked by one dot.**
*The wavelength of MODIS AOD is 550 nm. We added that to the figure caption and the related text. The dots mark the Grantley Adams International Airport on Barbados - this is already denoted in the caption of the figure.*

**(9) lines 93 - 96: You could drop the index 532 after clearly stating that all measurements have been performed at that wavelength.**
*We followed your suggestion and dropped the index in the revised manuscript for simplicity.*

**(10) line 98: Please clarify for the non-experts that DIAL gives the water vapour profiles and that HSRL gives the aerosol profiles.**
*We added a corresponding clarification.*

**(11) line 114: add reference to 10.1029/2009JD011862 and 10.1111/j.1600-0889.2011.00548.x regarding the use of lidar measurements to characterize aerosol mixtures**
*We added the citations and modified the references in the revised manuscript.*

**(12) line 143 and 145: AT these altitudes**
*We corrected the mistakes.*

**(13) Figure 5: please add the abbreviations for the different aerosol types (MA, DU, BB) in line 3 of the caption when marking their colour in the plot.**
*We added the abbreviations to the figure captions.*

**(14) line 211: dominates the aerosol mixture?**
*We followed your suggestion and changed the phrase.*

**(15) Figure 6: Is it possible to apply the information from Figure 5 to this plot to get more quantitative results rather than the coarse ellipses?**
*In this Figure, we explicitly wanted to separate the variables $\delta_p$ and $S$ with regard to $r_m$ from each other to give a better overview on the distribution of water vapor. This is why we used ellipses to roughly outline the regions of the two mixed regimes (BB-DU and MA-DU). We discussed how we could use the information of Figure 5 in the histograms shown in Figure 6, but thought it would be best to stick to the figures in the submitted manuscript. However, we decided to change the transparency of the ellipses, in a way that they don't appear too dominant.*

**Reply to Comments of Referee #2 on 11 March 2022**

**General comments**

**(I) First, I would like to underline that I consider the title to be appropriate and also, I like the presentation of the results in Fig.3. This way it is more visible the variability of the measurements during the three flights.**

*Thank for this comment. As the results of the study are based on measurements during one research campaign we agree with Referee #1 to rename the manuscript in a way that the reader immediately recognizes that this paper is on a lidar case study. Therefore, we changed the title accordingly (see response to comment (I) of Referee #1). We also followed the suggestion of Referee #1 to split up Figure 3 and Figure 4 such that the reader does not have to jump back and forth in the manuscript (see response to comment (IV) of Referee #1). This also increases the size of the panels in Figure 3.*

**(II) I agree with TROPOS team (comment by Albert Ansmann): The lidar scientists will use your measurements in follow-on papers, so please consider TROPOS suggestions. Furthermore, previous campaigns should be mentioned and also references to their publications.**

*Thank you for your feedback. For our response to this comment, please refer to the response to Albert Ansmann later in this document.*

**Minor technical suggestions /corrections**

**(1) Figure 1 should be moved in section 2.1 (after it is mentioned in the text). I think it should be kept in the manuscript; it gives the exact overlaying of the flights. Table 1 should be moved in section 3.1 (after it is mentioned in the text)**
*Thanks for these comments. Even though we placed Figure 1 and Table 1 after their introduction in the text (in the LATEXsource code), LATEXautomatically put them to the top of the pages. However, we managed to modify the template to put them to the bottom of the respective pages (after they have been introduced in the main body) in the revised manuscript. We followed your suggestion and kept Figure 1 in the revised manuscript as it gives a nice overview of all EUREC$^4$A flight tracks.*

**(2) Please specify in the caption what is the grey shaded areas on the right panels of Fig.3.**
*Thank you for this valuable comment. We specified that the median profiles of $R_{||}$ and $S$ refer to the grey shaded regions in the respective right panels.*

**Reply to Comments by Albert Ansmann on 11 March 2022**

(I) By reading the introduction (lines 30-36), the reader may get the impression, the authors introduce a new aspect: Winter transport of polluted dust over the remote tropical Atlantic towards, e.g., Barbados or even South America (Amazonia). But this impression needs to be avoided. We at TROPOS (partly together with Munich University, Wiegner, Gross, Freudenthaler) did so much work already in this field (since the SAMUM 2008 and later on in the framework the SALTRACE activities in 2013-2014) that needs to be mentioned:

Ansmann et al., Dust and smoke transport from Africa to South America: lidar profiling over Cape Verde and the Amazon rainforest, Geophys. Res. Lett., 36, L11802, doi:10.1029/2009GL037923, 2009.

Baars et al., Further evidence for significant smoke transport from Africa to Amazonia, Geophys. Res. Lett., 382, L20802, doi:10.1029/2011GL049200, 2011.

Rittmeister et al., Profiling of Saharan dust from the Caribbean to western Africa – Part 1: Layering structures and optical properties from shipborne polarization/Raman lidar observations, Atmos. Chem. Phys., 17, doi.org/10.5194/acp-17-12963-2017, 2017.

Ansmann et al., Profiling of Saharan dust from the Caribbean to western Africa – Part 2: Shipborne lidar measurements versus forecasts, Atmos. Chem. Phys., 17, 14987–15006, https://doi.org/10.5194/acp-17-14987-2017, 2017.

Haarig et al., ACP, 2019 (in the references)

Haarig et al., ACP, 2017, https://doi.org/10.5194/acp-17-14199-2017 on dry sea salt depolarization should also be mentioned as a source for uncertainties in the depolarization observations close to Barbados.

You may now realize why I personally was motivated to write this comment!

*Dear Albert,*

*thank you very much for this very valuable and important comment. It was never our intention to publish a paper in which we describe transatlantic winter-time transport of Saharan dust for the first time. Our aim was to investigate whether winter-time dust transport is also coming along with enhanced water vapor concentrations compared to the dry free troposphere (like already observed for summertime transport). This is why we did not focus on former studies on wintertime transport without focus on water vapor. However, we completely agree that those studies have to be mentioned in the manuscript. Otherwise the reader of the paper would get a false impression of the topic. Of course we included the list of papers that you have mentioned in the revised manuscript as they preceded our study. You can find the reworked parts in the appended marked-up manuscript version.*

(II) We have a severe problem with the PURE SMOKE particle linear depolarization ratio (PLDR) of 0.14 at 532nm in the troposphere! This has never been observed, except for the upper dry troposphere (for cases in which the smoke particles were unable to age quickly..., so that the irregular, fractal-like structures remained for a long time and caused enhanced PLDR values of up to 0.2, Burton et al.). However, in the lower and middle troposphere such enhanced PLDR values for pure smoke have never been observed. Extreme values may be here, 0.07 (Falcon observations during LACE98, and Falcon observation presented by Dahlkoetter et al., 2014, in the upper troposphere). But usually the smoke PLDR values are <0.05. This is the reason that one is able to properly separate smoke and dust contributions to lidar backscatter coefficients (Tesche et al., JGR2009, Tesche et al., Tellus2011). It is general accepted that aged biomass burning smoke particles at heights in the lower to middle troposphere cause PLDRs of <0.05. See Haarig et al., ACP 2018, on smoke in the troposphere and stratosphere... As long as you cannot demonstrate by lidar observations (or by a proper reference) that the PURE smoke PLDR is about 0.14 one should avoid to mention that. To our opinion, such a statement is not acceptable and even 'dangerous' because lidar scientists may use that in follow-on papers! All in all: Nice work!

*Thank you for this very helpful and valuable comment. You are absolutely right, that the depolarization ratios used for outlining the region of pure biomass burning aerosols in the submitted manuscript (i.e. Figure 6) are referring to values for non-aged and irregularly shaped biomass-burning aerosols. Of course we would have not observed such an aerosol species upstream the island of Barbados. During EUREC$^4$A we never observed pure*

*biomass burning aerosol. We always observed aerosol mixtures of biomass burning aerosol and mineral dust. The optical properties of these aerosol mixtures are also in good agreement with those observed during SAMUM-2 at the beginning of transatlantic transport (Tesche et al. 2009, in the references). Furthermore, we rectified this mistake and changed the assumed particle linear depolarization ratio at 532 nm as well as the lidar ratio of biomass burning aerosol to values published by Haarig et al. 2018 for the lidar-based aerosol separation. This shifts the reference lines in Figure 6, but has no further impact on our results.*

[revised manuscript text omitted]

with the coefficients $\mathrm{D_A}$ and $\mathrm{D_B}$,

$$\mathrm{D_A} := \frac{\delta_{\mathrm{p(532),DU}} - \delta_{\mathrm{p(532)}}}{\mathrm{S_{DU}}(1 + \delta_{\mathrm{p(532),DU}})};$$

$$\mathrm{D_B} := \frac{\delta_{\mathrm{p(532)}} - \delta_{\mathrm{p(532),BB|MA}}}{\mathrm{S_{BB|MA}}(1 + \delta_{\mathrm{p(532),BB|MA}})}.$$

Here, $S_{DU}$ and $S_{BB|MA}$ are the known lidar ratios of Saharan mineral dust (50±4) as well as of biomass burning aerosols (63±7) and marine aerosol (18±5) (mean values of observations by Groß et al., 2013). $\delta_{p(532),DU}$ and $\delta_{p(532),BB|MA}$ are the corresponding known particle linear depolarization ratios (Saharan mineral dust: 27±2 %; biomass burning aerosol: 14±2 %; marine aerosol: 3±1 %). For $S$ the percentage is calculated using,

$$[\%_\mathrm{S}] = \frac{\mathrm{S_{BB|MA}}(\mathrm{S_{DU}} - \mathrm{S})}{\mathrm{S}(\mathrm{S_{BB|MA}} - \mathrm{S_{DU}})}.$$

[revised manuscript text omitted]